

# Vocal correlates of individual sooty mangabey travel speed and direction

Christof Neumann[1] and Klaus Zuberbühler[1,2]

[1] Department of Comparative Cognition, Institute of Biology, Université de Neuchâtel, Neuchâtel, Switzerland
[2] School of Psychology & Neurosciences, University of St Andrews, St Andrews, Fife, United Kingdom

## ABSTRACT

Many group-living animals coordinate movements with acoustic signals, but so far most studies have focused on how group movements are initiated. In this study, we investigated movement patterns of wild sooty mangabeys (*Cercocebus atys*), a mostly terrestrial, forest-dwelling primate. We provide quantitative results showing that vocalization rates of mangabey subgroups, but not of focal individuals, correlated with focal individuals' current movement patterns. More interestingly, vocal behaviour predicted whether individuals changed future speed, and possibly future travel direction. The role of vocalizations as a potential mechanism for the regulation of group movement was further highlighted by interaction effects that include subgroup size and the quality of poly-specific associations. Collectively, our results suggest that primate vocal behaviour can function beyond travel initiation in coordination and regulation of group movements.

## INTRODUCTION

Living in groups can convey considerable benefits for individuals, such as increased predator detection and foraging success. Nevertheless, group living incurs costs; for example, through increased competition among individuals for resources or exposure to diseases (*Elgar, 1989*; *Sterck, Watts & van Schaik, 1997*; *Krause & Ruxton, 2002*; *Altizer et al., 2003*; *Majolo, de Bortoli Vizioli & Schino, 2008*). To benefit from the presence of others, individuals need to regulate group cohesion, especially during travel, and signals such as vocalizations are likely to play a key role in this context (*Boinski & Garber, 2000*; *Conradt & Roper, 2005*; *da Cunha & Byrne, 2009*; *Petit & Bon, 2010*; *Fischer & Zinner, 2011a*; *Fischer & Zinner, 2011b*).

Most previous research in group coordination has focused on how movements are initiated (e.g., *Stewart & Harcourt, 1994*; *Radford, 2004*; *Bousquet, Sumpter & Manser, 2011*), which has revealed a variety of mechanisms (*Conradt & Roper, 2005*; *Petit & Bon, 2010*). In contrast, we are not aware of empirical work that has looked at the role of vocalizations during travel, such as how speed and changes in direction are determined, and how movements are terminated. As such, initiation and termination of group movements may only represent the extreme ends of a more complex phenomenon, which

Corresponding author
Christof Neumann,
christofneumann1@gmail.com

may require additional communicative mechanisms that are used to regulate movement patterns along the way.

In addition, researchers working on movement initiations normally study discrete bouts of movements, such as when travelling from a resting place to a food resource. In such situations typically one or several leader individuals are followed or a group consensus is reached, through the production of specific vocal and other behavioural signals (e.g., *Black, 1988*; *Fletcher, 2007*; *Bousquet, Sumpter & Manser, 2011*). It is possible that such discrete and conspicuous movement bouts are typical only for species that exploit clumped food or water resources (e.g., *Asensio et al., 2011*; *Noser & Byrne, 2014*). Mechanisms for species foraging on relatively evenly distributed food resources may be different, especially if foraging involves more or less continuous travel.

Vocalizations are particularly well studied with respect to their role in initiating movements (e.g., *Poole et al., 1988*; *Bousquet, Sumpter & Manser, 2011*; *Fischer & Zinner, 2011a*), perhaps because acoustic signals can bridge the continuum between local and global information of individuals within a group much better than visual signals. The latter can function only in the local domain (*Conradt & Roper, 2005*; *Petit & Bon, 2010*, see also *Couzin et al. (2005)* and *Strandburg-Peshkin et al. (2015)*) and the use of which is therefore constrained by habitat characteristics. Thus, vocalizations might be particularly well suited for the propagation of local information through large groups, especially if individuals are spread beyond an individual's visual range in dense habitats.

A range of studies have suggested that some vocalizations function in the context of maintaining group cohesion, though without addressing the specific mechanisms by which cohesion is regulated on an individual level (e.g., *Janik & Slater, 1998*; *da Cunha & Byrne, 2009*; *Mumm, Urrutia & Knörnschild, 2014*). However, simply demonstrating a relationship between vocal rates and behavioural markers of cohesion falls short of elucidating the mechanisms by which such signals function in group cohesion (*Palombit, 1992*; *da Cunha & Byrne, 2009*). In our opinion, it is crucial to address *changes* in addition to states of individual behaviour, i.e., investigating the temporal dynamics of movement behaviour. This approach highlights the spatio-temporal quality of how group cohesion can be achieved by individuals moving in space and time.

In this study, we investigated to what extent vocal behaviour plays a role in regulating individual movements during travel, beyond travel initiations. Crucially, we addressed not only the relationship between current vocal behaviour and current movement patterns, but also whether current vocal behaviour predicted future movement, i.e., changes in movement. We conducted our study on sooty mangabeys (*Cercocebus atys*), a forest dwelling, terrestrial primate species that lives in large groups of up to 100 individuals (*McGraw, Zuberbühler & Noë, 2007*). Sooty mangabey foraging behaviour mainly consists of individuals searching for edible items on the forest floor (*McGraw, Zuberbühler & Noë, 2007*). As a result, their movements are rarely in the form of distinct travel bouts from one resource to the next but are characterised by continuous movements with variable speed in a general direction. Within this general direction, individuals follow their own paths, which often differ from the general direction of the group.

Our focus in this study is on describing these spatio-temporal movement patterns during foraging and relate them to vocal production. Particularly, we studied the relationships between individual movement speed and direction and (1) vocal rates of focal animals and (2) vocal rates of individuals in the focal animal's immediate vicinity (subgroup). We focused on the two most common mangabey vocalizations, grunts and twitters, whose exact communicative functions are currently unknown, although they are predominantly given during foraging (*Range & Fischer, 2004*). Preliminary observations have suggested that both call types play a role in group coordination. All other vocalizations were also recorded but then pooled into an 'other vocalizations' category.

Sooty mangabeys frequently form associations with other, arboreal primate species, mostly Diana monkeys, *Cercopithecus diana* and red colobus, *Procolobus badius*, (*McGraw, Zuberbühler & Noë, 2007*), presumably to increase predator detection (*Bergmüller, 1998*; *McGraw & Bshary, 2002*; *Heymann, 2011*), which enabled us to test whether movement patterns were further influenced by vocal rates of other species in addition to their mere presence.

## METHODS

We observed adult individuals in a free-ranging, habituated group of sooty mangabeys of approximately 90 individuals at Taï National Park in Côte d'Ivoire (*Range & Fischer, 2004*; *Janmaat, Byrne & Zuberbühler, 2006*; *McGraw, Zuberbühler & Noë, 2007*). Our study group has been subject to other research projects in the past and all adult individuals are identifiable based on physical characteristics. During focal animal follows, we used a combination of all-occurrence and instantaneous sampling (*Altmann, 1974*). Data on the focal individual's vocalizations (grunts, twitters, other vocalizations) were collected continuously. In two-minute intervals, we noted the behaviour of the focal individuals and the number of individuals present within 10 m. We refer to the individuals within 10 m as the focal individual's subgroup, which for the purpose of our study is considered a proxy for the focal animal's immediate social environment. Subgroup size ranged from 0 to 17 individuals (median = 4), excluding the focal individual. We did not record the age and sex composition of the subgroup.

Every 10 min, we noted the number of other primate species associated with the mangabey group. Another species was associated if we detected the presence of at least one individual of another species within 50 m from the focal individual (*McGraw & Bshary, 2002*). GPS coordinates were recorded automatically every 30 s with a Garmin Rhino 650 unit. We continuously recorded the soundscape around the focal individual (Sennheiser MKH-416 microphone, Marantz PMD660 recorder). From these audio recordings, we counted the number of monkey vocalizations audible to the coder and assigned them to either mangabeys (distinguishing grunts, twitters and others) or any of the associated primate species to calculate rates of vocalizations. Classification of vocalizations was done by ear from the audio recordings, based on descriptions in *Range & Fischer (2004)* and with the help of experienced local field assistants, who have worked in the study area for years. Vocalizations of the focal individuals were tagged during focal follows by giving spoken comments onto the recording. Rates of mangabey

subgroup vocalizations in the soundscape were positively correlated with subgroup size (grunt: $r = 0.22$; twitter: $r = 0.20$; other: $r = 0.34$), but model diagnostics suggested that this was not problematic (see ESM). Hence, we refer to these vocalization rates in the soundscape as subgroup grunts, subgroup twitters and subgroup other vocalizations.

We used linear mixed models to address our questions. We created time-blocks of five minutes, for which we established the distance covered by the focal animal as response variable (numeric, hereafter: speed, i.e., distance covered per 5-min time-block). As our major predictor variables of interest we used whether or not the focal animal produced at least one grunt or other vocalization (binary), vocal rates of mangabeys as audible in the soundscape around the focal individual (grunts, twitters and other vocalizations, excluding the focal animal's vocalizations, all numeric) and the rate of primate, non-mangabey vocalizations (numeric). In addition, we controlled for a number of variables that might influence movement speed, i.e., the average number of individuals within 10 m of the focal animal (numeric, hereafter: subgroup size), the sex of the focal individual (binary), and the number of associated primate species (numeric). Vocalizations of focal animals were too infrequent to allow calculating meaningful calling rates and hence were coded binary. For the same reason, we could not include twitter production of focal individuals as a predictor variable as no focal animal produced a twitter during the focal follows. We did not consider behaviour/activity as variable in our models, because pure travel behaviour was rare (2.3% of activity budget, as opposed to feeding, foraging, resting and socialising, see ESM for details). We incorporated an auto-correlation term to control for temporal dependence of data points (*Fürtbauer et al., 2011*) and fitted individual ID and calendar date as random intercepts.

Change points, i.e. points in space and time at which individuals modified the general direction into which they moved, were assessed following procedures described by *Byrne et al. (2009)*. In brief, the change point test decomposes an individual track into smaller segments and examines whether a given track segment is aligned with systematically varied numbers of segments before and after it (*Byrne et al., 2009*). A more detailed description of the method can be found in *Byrne et al. (2009)* and examples of its application are *Asensio et al. (2011)*, *Janmaat, Ban & Boesch (2013)* and *Noser & Byrne (2014)*. We used the following parameters to calculate change points: $q = 6$, $\alpha = 0.05$, $N = 1,000$ and a tolerance of 0.00002.

Our modelling approach was two-fold. First, we aimed to describe the co-variation between speed and the predictor variables within the same 5-min time blocks. Second, within a given time block we used our variables to predict speed in the following time block, i.e. future speed, while controlling for current speed. We followed the same approach for modelling probabilities of direction changes (presence or absence of 'change points') in current and future time blocks. Table 1 summaries the design. Our sample comprised 16 individuals (11 females, 5 males), encompassing 175 5-min data points totalling 14.6 h of focal observations (range: 0.25–1.75 h per individual).

All models were built in R 3.1.1 (*R Core Team, 2014*) with the lmer and glmer functions in the lme4 package (v. 1.1.11, *Bates et al., 2015*). Statistical significance was established using likelihood ratio tests (LRT, *Dobson, 2002*) comparing full models with their

**Table 1 Outline of analysis strategy and summary of results.** We built four models that tested variation in current and future speed and probability of direction changes of sooty mangabeys. Results in the table represent comparisons of full versus null models using likelihood ratio tests.

|  | Speed | Change points |
| --- | --- | --- |
| Current time block | $\chi^2_{14} = 36.59$ | $\chi^2_{14} = 20.35$ |
|  | $p = 0.0009$ | $p = 0.1194$ |
| Future time block | $\chi^2_{14} = 26.60$ | $\chi^2_{14} = 23.36$ |
|  | $p = 0.0217$ | $p = 0.0546$ |

respective null model. These null models contained the same random effects as the full models and sex as fixed effect. Depending on the model, we also included the autocorrelation term and speed or direction change in the previous time block as additional terms in the null models. $R^2$ values were computed following *Nakagawa & Schielzeth (2013)* using the MuMIn package (v.1.15.6, *Bartoń, 2016*). We tested several two-way interactions in each model, which were retained only if they improved model fit as determined by LRTs and were otherwise removed to allow interpretation of main effects (*Mundry, 2011*; *Hector, von Felten & Schmid, 2010*). Specifically, we included interactions between subgroup size and vocalizations of mangabeys (both focal individuals' (grunt, other) and subgroups' (grunt, twitter, other)), reasoning that effects of vocalizations may differ according to the number of individuals in proximity of the focal individual. We also included the two-way interaction between number of associated species and calling rate of associated species. We report and interpret results only for the major test predictor variables concerning vocal behaviour. Where appropriate, this includes reporting of interactions. More details on methods, analyses and checks of model assumptions can be found in the ESM.

This study was entirely observational and adhered to the legal requirements of Côte d'Ivoire and Switzerland, as well as to the Animal Behavior Society Guidelines for the Use of Animals in Research. Research permissions were granted by the Ministère de la Recherche Scientifique et Technique of Côte d'Ivoire.

## RESULTS

### Travel speed

Both models for current and future travel speed were different from their respective null models (current: $\chi^2_{14} = 36.59$, $p = 0.0009$, $R^2_m = 0.36$; future: $\chi^2_{14} = 26.60$, $p = 0.0217$, $R^2_m = 0.34$; LRT; Table 1; full model results in Tables 2 and 3).

#### Current travel speed

We found no strong relationships between vocalizations of focal animals and their current travel speed (grunts: $\beta \pm se = -0.142 \pm 0.107$, $t = -1.332$; other vocalizations: $\beta \pm se = -0.049 \pm 0.132$, $t = -0.372$; twitters: not tested, see methods).

In contrast, high rates of twitters in the soundscape (subgroup twitters) were associated with low current speed of the focal individual ($\beta \pm se = -0.108 \pm 0.051$, $t = -2.146$, Fig. 1). We found no such significant effect for subgroup grunts ($\beta \pm se = -0.068 \pm 0.049$,

**Table 2 Results of LMM testing variation in *current speed*.** Given are results for the full model, including all interactions, and of the final model, from which non-significant interaction terms were removed. Significance of interpretable terms in the final model (interaction terms and main effects of terms not included in an interaction) was assessed with likelihood ratio tests.

| | Full model | | Final model | | LRT | |
|---|---|---|---|---|---|---|
| | $\beta \pm se$ | $t$ | $\beta \pm se$ | $t$ | $\chi^2_1$ | $p$ |
| Intercept | −0.35 ± 1.27 | −0.28 | −0.31 ± 1.27 | −0.25 | | |
| Subgroup size | −0.07 ± 0.05 | −1.32 | −0.05 ± 0.05 | −0.99 | | |
| Focal grunt (yes) | −0.15 ± 0.11 | −1.44 | −0.14 ± 0.11 | −1.33 | 1.76 | 0.1844 |
| Focal other vocalization (yes) | −0.03 ± 0.13 | −0.19 | −0.05 ± 0.13 | −0.37 | 0.14 | 0.7107 |
| Subgroup grunt | −0.07 ± 0.05 | −1.44 | −0.07 ± 0.05 | −1.38 | 1.90 | 0.1680 |
| Subgroup other | 0.11 ± 0.05 | 2.45 | 0.10 ± 0.05 | 2.16 | | |
| Subgroup twitter | −0.12 ± 0.05 | −2.28 | −0.11 ± 0.05 | −2.15 | 4.53 | 0.0333 |
| Number of associated species | −0.03 ± 0.08 | −0.43 | −0.00 ± 0.08 | −0.06 | | |
| Vocal rate of other species | −0.01 ± 0.05 | −0.19 | −0.03 ± 0.05 | −0.53 | | |
| Sex (male) | 4.03 ± 2.27 | 1.78 | 3.99 ± 2.27 | 1.75 | 2.87 | 0.0901 |
| Auto-correlation | −2.60 ± 0.14 | −18.53 | −2.61 ± 0.14 | −18.48 | 113.99 | 0.0000 |
| IA subgroup size: focal grunt | 0.19 ± 0.11 | 1.68 | | | | |
| IA subgroup size: focal other | −0.23 ± 0.18 | −1.29 | | | | |
| IA subgroup size: subgroup grunt | 0.03 ± 0.05 | 0.56 | | | | |
| IA subgroup size: subgroup other | −0.16 ± 0.04 | −3.66 | −0.14 ± 0.04 | −3.41 | 11.18 | 0.0008 |
| IA subgroup size: subgroup twitter | 0.03 ± 0.05 | 0.74 | | | | |
| IA associated species: vocal rate of other species | 0.10 ± 0.05 | 2.02 | 0.10 ± 0.05 | 2.02 | 4.01 | 0.0453 |

**Note:**
Reference levels of categorical variables are: focal grunt = 'no,' focal other = 'no,' and sex = 'female.' Test levels are given in parentheses. LRT, likelihood ratio test, IA, interaction.

$t = −1.383$). For other subgroup vocalizations the effect of calling rate was mediated by subgroup size (interaction: $\beta \pm se = −0.140 \pm 0.041$, $t = −3.410$, Fig. 2), insofar as in smaller subgroups, higher call rates were associated with higher current speed, while the opposite was the case for larger subgroups.

Finally, the calling rate of other primate species also influenced current speed and this was mediated by the number of associated species ($\beta \pm se = 0.101 \pm 0.050$, $t = 2.021$): with fewer associated species, higher call rates were associated with lower current speed, while the opposite was the case for larger poly-specific groups consisting of many species (Fig. 3).

### Future travel speed

As with current speed, we found no statistically significant effects of focal animal vocalizations on future travel speed (grunts: $\beta \pm se = −0.107 \pm 0.143$, $t = −0.744$; other vocalizations: $\beta \pm se = −0.280 \pm 0.169$, $t = −1.652$; twitters: not tested, see methods).

Focal individuals decreased future speed (i.e. travelled slower in the future) if the subgroup produced more other vocalizations ($\beta \pm se = −0.154 \pm 0.062$, $t = −2.497$, Fig. 4). Subgroup grunt rate was also associated with future speed, but this relationship depended on subgroup size (interaction: $\beta \pm se = −0.130 \pm 0.055$, $t = −2.349$, Fig. 5): in small subgroups, individuals travelled faster in the future if the subgroup produced

**Table 3 Results of LMM testing variation in _future speed_.** Given are results for the full model, including all interactions, and of the final model, from which non-significant interaction terms were removed. Significance of interpretable terms in the final model (interaction terms and main effects of terms not included in an interaction) was assessed with likelihood ratio tests.

| | Full model | | Final model | | LRT | |
|---|---|---|---|---|---|---|
| | $\beta \pm$ se | $t$ | $\beta \pm$ se | $t$ | $\chi^2_1$ | $p$ |
| Intercept | 0.16 ± 0.71 | 0.22 | 0.20 ± 0.69 | 0.28 | | |
| Subgroup size | −0.01 ± 0.07 | −0.12 | −0.03 ± 0.06 | −0.53 | | |
| Focal grunt (yes) | −0.15 ± 0.14 | −1.05 | −0.11 ± 0.14 | −0.74 | 0.55 | 0.4575 |
| Focal other vocalization (yes) | −0.31 ± 0.17 | −1.77 | −0.28 ± 0.17 | −1.65 | 2.70 | 0.1007 |
| Subgroup grunt | 0.16 ± 0.06 | 2.63 | 0.15 ± 0.06 | 2.36 | | |
| Subgroup other | −0.15 ± 0.06 | −2.45 | −0.15 ± 0.06 | −2.50 | 6.06 | 0.0139 |
| Subgroup twitter | −0.02 ± 0.07 | −0.32 | −0.03 ± 0.06 | −0.41 | 0.16 | 0.6855 |
| Number of associated species | −0.05 ± 0.10 | −0.49 | −0.05 ± 0.10 | −0.49 | 0.22 | 0.6354 |
| Vocal rate of other species | 0.01 ± 0.07 | 0.07 | −0.01 ± 0.07 | −0.21 | 0.04 | 0.8378 |
| Sex (male) | 0.67 ± 1.26 | 0.53 | 0.52 ± 1.24 | 0.42 | 0.18 | 0.6747 |
| Control speed | 0.00 ± 0.06 | 0.02 | 0.02 ± 0.06 | 0.33 | 0.10 | 0.7461 |
| Auto-correlation | −1.60 ± 0.13 | −12.76 | −1.60 ± 0.13 | −12.58 | 61.20 | 0.0000 |
| IA subgroup size: focal grunt | −0.12 ± 0.15 | −0.79 | | | | |
| IA subgroup size: focal other | 0.00 ± 0.23 | 0.01 | | | | |
| IA subgroup size: subgroup grunt | −0.09 ± 0.06 | −1.41 | −0.13 ± 0.06 | −2.35 | 5.39 | 0.0203 |
| IA subgroup size: subgroup other | −0.10 ± 0.06 | −1.70 | | | | |
| IA subgroup size: subgroup twitter | −0.02 ± 0.06 | −0.33 | | | | |
| IA associated species: vocal rate of other species | 0.08 ± 0.06 | 1.37 | | | | |

**Note:**
Reference levels of categorical variables are: focal grunt = 'no,' focal other = 'no,' and sex = 'female.' Test levels are given in parentheses. LRT, likelihood ratio test, IA, interaction.

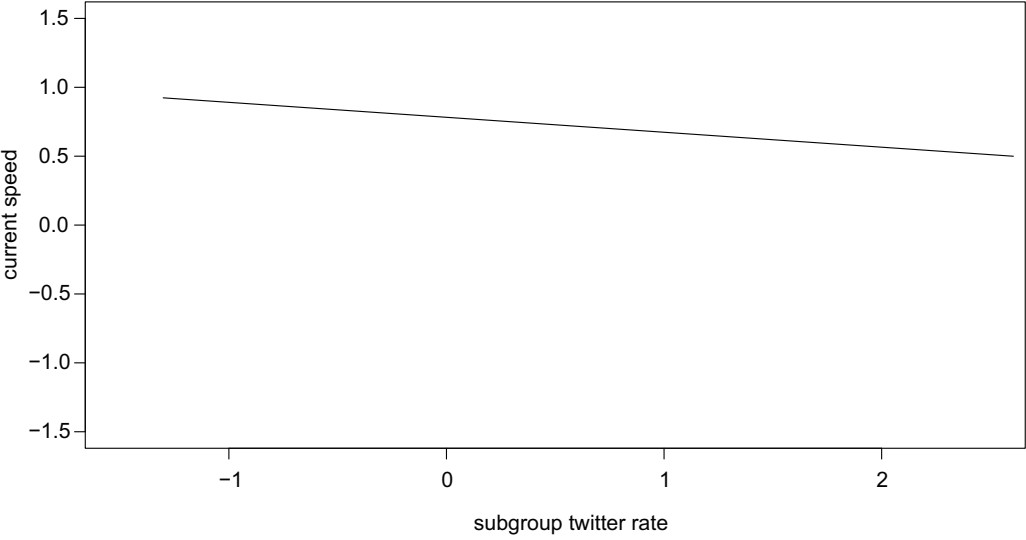

**Figure 1 Higher rates of subgroup twitters were associated with lower travel speed.**

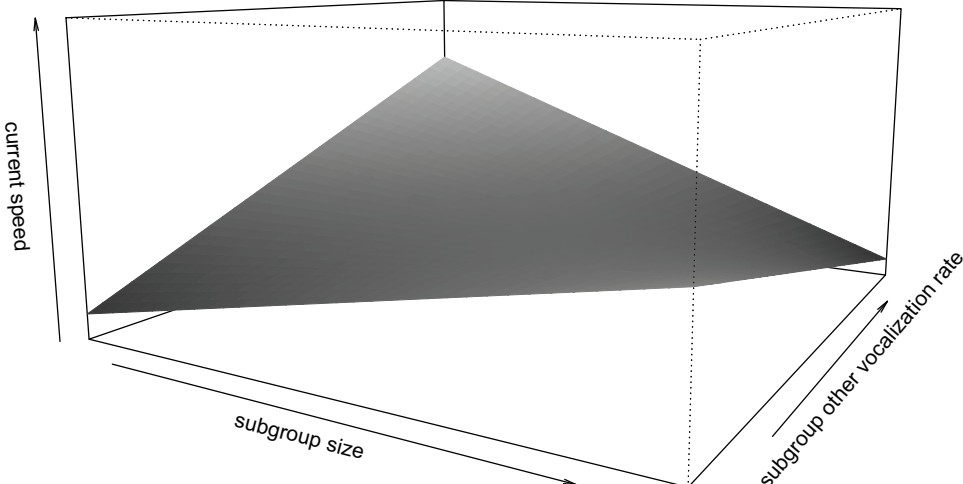

**Figure 2 Interaction effect of subgroup size and subgroup rate of other vocalizations.** In smaller subgroups, higher calling rates were associated with faster travel speed. In larger subgroups, lower calling rates were associated with higher travel speed of focal individuals. Speed (along the z-axis) ranges between −1.5 and 1.5. Limits along the x- and y-axes correspond to the range of standardized values in the data.

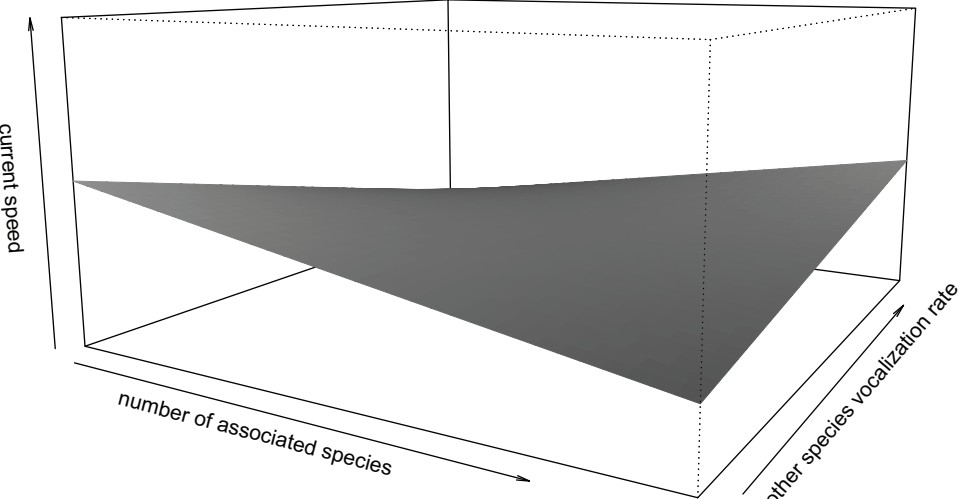

**Figure 3 Interaction effect of number of associated species and calling rate of other species.** With fewer associated species, higher calling rates corresponded to lower travel speed. With more associated species, higher calling rates corresponded to higher speed of focal individuals. Speed (along the z-axis) ranges between −1.5 and 1.5. Limits along the x- and y-axes correspond to the range of standardized values in the data.

grunts at higher rates, while the opposite was found for large subgroups. There was no statistically significant effect of subgroup twitter rate on future speed ($\beta \pm se = -0.026 \pm 0.064$, $t = -0.406$).

The calling rate of other primate species appeared to have no pronounced effect on future travel speed of focal individuals ($\beta \pm se = -0.014 \pm 0.068$, $t = -0.208$).

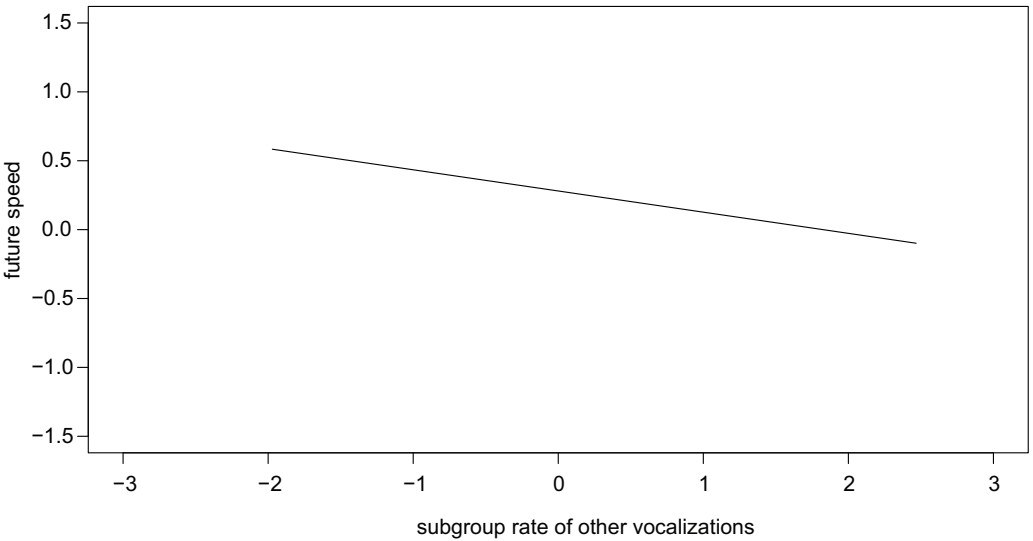

**Figure 4 Individuals slowed down in the future if the subgroup produced other vocalizations at higher rates.**

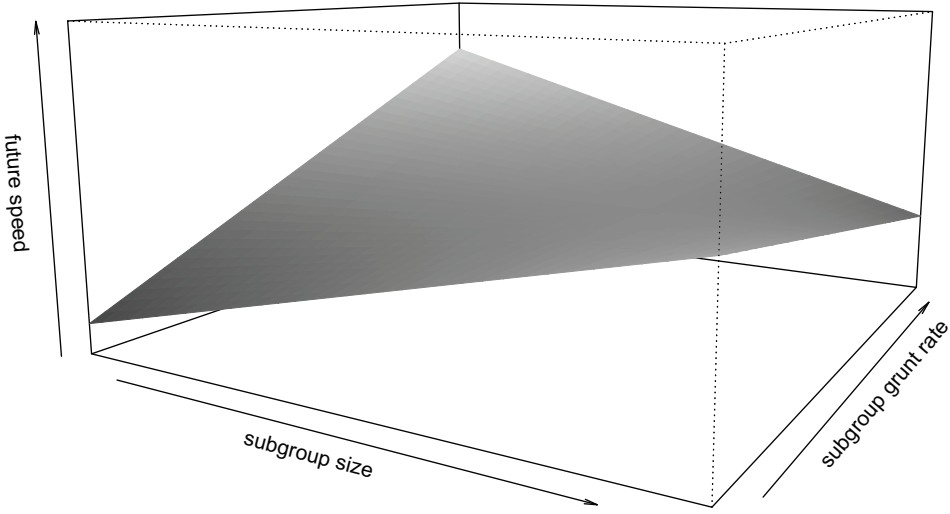

**Figure 5 Interaction between subgroup size and subgroup grunt rate and its effect on future speed of individual mangabeys.** In smaller subgroups, individuals increased future speed with higher subgroup grunt rates. In larger subgroups, individuals decreased future speed with higher subgroup grunt rates. Speed (along the z-axis) ranges between −1.5 and 1.5. Limits along the x- and y-axes correspond to the range of standardized values in the data.

## Direction changes

Regarding changes in direction, neither the 'current direction changes' nor the 'future direction changes' full model was significant at $\alpha = 0.05$ (current: $\chi^2_{14} = 20.35$, $p = 0.1194$, $R^2_m = 0.28$; future: $\chi^2_{14} = 23.36$, $p = 0.0546$, $R^2_m = 0.30$; Table 1; full model results in Tables 4 and 5). Given the low $p$ value of the future model, we continued to explore this model.

**Table 4  Results of GLMM testing variation in *current direction changes*.** Given are results for the full model, including all interactions. Since the overall model was not statistically significant at $\alpha = 0.05$ no final model or tests of individual terms are presented.

| | $\beta \pm$ se | $z$ |
|---|---|---|
| Intercept | $-1.68 \pm 0.34$ | $-4.89$ |
| Subgroup size | $-0.78 \pm 0.31$ | $-2.54$ |
| Focal grunt (yes) | $-0.20 \pm 0.49$ | $-0.41$ |
| Focal other vocalization (yes) | $0.97 \pm 0.65$ | $1.49$ |
| Subgroup grunt | $0.01 \pm 0.26$ | $0.05$ |
| Subgroup other | $0.02 \pm 0.25$ | $0.07$ |
| Subgroup twitter | $0.14 \pm 0.27$ | $0.51$ |
| Number of associated species | $-0.14 \pm 0.31$ | $-0.45$ |
| Vocal rate of other species | $0.32 \pm 0.27$ | $1.19$ |
| Sex (male) | $0.05 \pm 0.61$ | $0.08$ |
| IA subgroup size: focal grunt | $0.22 \pm 0.60$ | $0.36$ |
| IA subgroup size: focal other | $2.34 \pm 1.04$ | $2.24$ |
| IA subgroup size: subgroup grunt | $0.04 \pm 0.30$ | $0.12$ |
| IA subgroup size: subgroup other | $-0.56 \pm 0.28$ | $-1.98$ |
| IA subgroup size: subgroup twitter | $-0.08 \pm 0.30$ | $-0.26$ |
| IA associated species: vocal rate of other species | $0.36 \pm 0.20$ | $1.77$ |

**Note:**
Reference levels of categorical variables are: focal grunt = 'no,' focal other = 'no,' and sex = 'female.' Test levels are given in parentheses. IA, interaction.

**Table 5  Results of GLMM testing variation in *future direction changes*.** Given are results for the full model, including all interactions, and of the final model, from which non-significant interaction terms were removed. Significance of interpretable terms in the final model (interaction terms and main effects of terms not included in an interaction) was assessed with likelihood ratio tests.

| | Full model | | Final model | | LRT | |
|---|---|---|---|---|---|---|
| | $\beta \pm$ se | $z$ | $\beta \pm$ se | $z$ | $\chi^2_1$ | $p$ |
| Intercept | $-1.80 \pm 0.38$ | $-4.72$ | $-1.76 \pm 0.36$ | $-4.85$ | | |
| Subgroup size | $-0.30 \pm 0.33$ | $-0.91$ | $-0.43 \pm 0.25$ | $-1.69$ | | |
| Focal grunt (yes) | $0.26 \pm 0.52$ | $0.51$ | $0.46 \pm 0.48$ | $0.95$ | $0.89$ | $0.3444$ |
| Focal other vocalization (yes) | $0.79 \pm 0.72$ | $1.10$ | $0.50 \pm 0.63$ | $0.79$ | $0.61$ | $0.4367$ |
| Subgroup grunt | $-0.03 \pm 0.28$ | $-0.11$ | $-0.10 \pm 0.24$ | $-0.40$ | $0.16$ | $0.6917$ |
| Subgroup other | $-0.40 \pm 0.27$ | $-1.48$ | $-0.32 \pm 0.25$ | $-1.30$ | | |
| Subgroup twitter | $0.51 \pm 0.30$ | $1.69$ | $0.49 \pm 0.25$ | $1.94$ | $3.86$ | $0.0493$ |
| Number of associated species | $-0.21 \pm 0.28$ | $-0.75$ | $-0.31 \pm 0.25$ | $-1.22$ | $1.48$ | $0.2238$ |
| Vocal rate of other species | $0.15 \pm 0.28$ | $0.54$ | $-0.03 \pm 0.23$ | $-0.12$ | $0.01$ | $0.9064$ |
| Sex (male) | $0.53 \pm 0.63$ | $0.84$ | $0.71 \pm 0.61$ | $1.17$ | $1.36$ | $0.2437$ |
| Control change point | $1.19 \pm 0.51$ | $2.33$ | $1.31 \pm 0.48$ | $2.76$ | $7.50$ | $0.0062$ |
| IA subgroup size: focal grunt | $-1.25 \pm 0.68$ | $-1.84$ | | | | |
| IA subgroup size: focal other | $1.46 \pm 0.93$ | $1.58$ | | | | |
| IA subgroup size: subgroup grunt | $0.17 \pm 0.28$ | $0.61$ | | | | |
| IA subgroup size: subgroup other | $-0.79 \pm 0.34$ | $-2.28$ | $-0.56 \pm 0.29$ | $-1.94$ | $4.44$ | $0.0352$ |
| IA subgroup size: subgroup twitter | $0.04 \pm 0.33$ | $0.11$ | | | | |
| IA associated species: vocal rate of other species | $0.33 \pm 0.21$ | $1.57$ | | | | |

**Note:**
Reference levels of categorical variables are: focal grunt = 'no,' focal other = 'no,' and sex = 'female.' LRT, likelihood ratio test, IA, interaction.

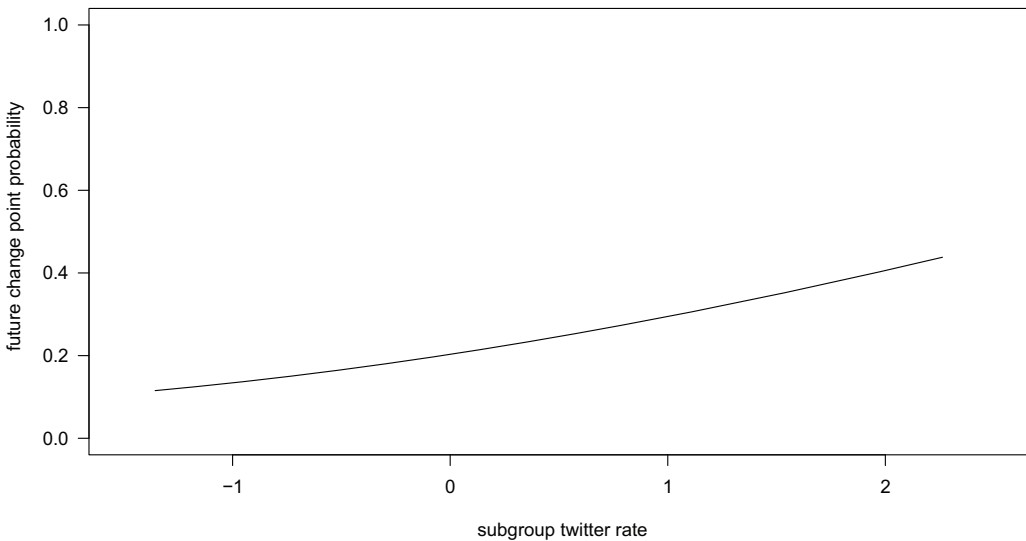

**Figure 6 Individuals were more likely to change direction if the subgroup twitter rate was higher.**

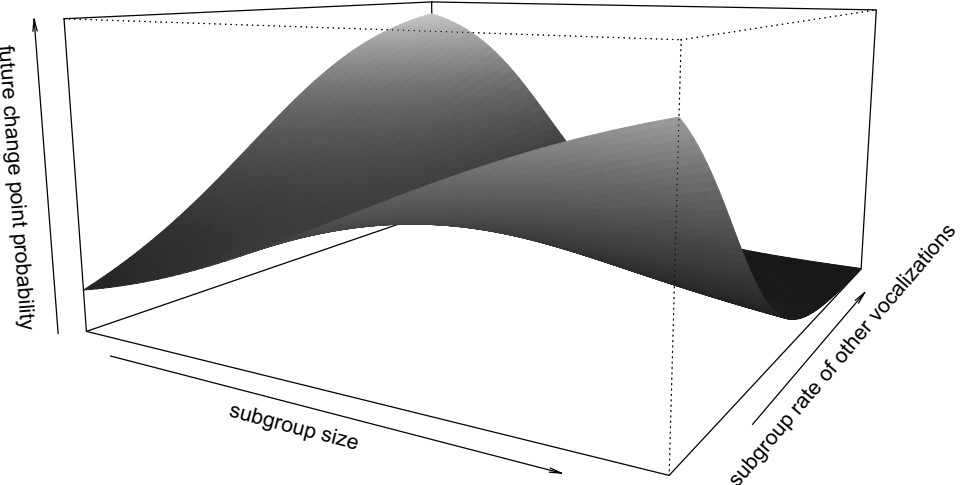

**Figure 7 Interaction between subgroup size and subgroup rate of other vocalizations.** Individuals were more likely to change direction in the future if the subgroup was small but vocalized at high rates or if subgroup size was large but produced little other vocalizations. Probability of a change in direction (along the z-axis) ranges between 0 and 1. Limits along the x- and y-axes correspond to the range of standardized values in the data.

### Future direction changes

Focal animals' vocalizations did not significantly predict the probability of a change in future travel direction (grunts: $\beta \pm se = 0.458 \pm 0.482$, $z = 0.950$; other vocalizations: $\beta \pm se = 0.499 \pm 0.635$, $z = 0.786$; twitters: not tested, see methods).

Individuals were more likely to change direction in the future if the rate of twitters in the subgroup increased ($\beta \pm se = 0.494 \pm 0.255$, $z = 1.941$, Fig. 6). The probability of an individual changing direction in the future also depended on the interaction between

subgroup size and rate of other vocalizations in the subgroup (interaction: $\beta \pm se = -0.559 \pm 0.288$, $z = -1.944$, Fig. 7). In smaller subgroups, future direction changes were more likely with high rates of vocalizations compared to low vocalization rates. In larger subgroups, this pattern is reversed, such that direction changes in the future were more likely with low vocalization rates compared to high vocalization rates. There was no statistically significant effect of subgroup grunt rate on the probability of changing direction in the future ($\beta \pm se = -0.095 \pm 0.241$, $t = -0.396$).

As in the model of future speed, the calling rate of other primate species appeared to not have a pronounced effect on the probability of focal individuals changing direction in the future (calling rate of other species: $\beta \pm se = -0.027 \pm 0.228$, $t = -0.118$).

## DISCUSSION

Our results indicate that the travel speed and changes in direction of focal individuals co-varied with complex interactions of conspecific and heterospecific vocalizations. Notably, our results indicate that individual movement patterns were largely independent of the focal animal's own vocal behaviour. In contrast, we found effects of the subgroup's collective vocal behaviour and vocal rates of associated primate species, which were related to individual movement patterns.

Interestingly, differences in vocal rates were often not directly linked with differences in travel speed and direction changes, but were mediated by the social environment, i.e. subgroup size. The only vocal predictor of future speed that was independent of social factors was how many 'other' vocalizations were produced in the subgroup, which generally slowed down individuals. Most likely, this was a consequence of high rates of vocalizations produced during important social interactions, such as aggression and mating, which tend to take place while animals remain stationary.

Our results also suggest that the rate of twitters in the soundscape around the focal animal predicted whether or not this individual changed direction: individuals were more likely to change direction if twitters were more frequent as compared to when twitters were rare. Whether or not these direction changes led individuals towards the source of twitters is hitherto unknown and we do not have information about where these twitters originated from the focal individual's perspective. Given the proposed function of twitters in foraging (*Range & Fischer, 2004*), it would be interesting to see whether these calls serve as food calls similar to those described in chimpanzees (*Pan troglodytes*), for example (*Schel et al., 2013*). In chimpanzees, it is thought that such food calls attract individuals to valuable food sources, presumably as a means to establish or maintain social bonds with group members (*Schel et al. (2013)*, see also *Clay, Smith & Blumstein (2012)* for a review on alternative functions of food calls).

Similar to our results on future speed, we found that the effect of the subgroup's 'other vocalizations' rate on the probability of changing future direction was modulated by subgroup size. As with changes in speed, it is most likely that these effects are a consequence of relevant social interactions nearby, for which 'other vocalizations' may be indicators and which may subsequently trigger changes in direction of focal individuals.

Note, however, that our results on direction changes did not reach the conventional level of statistical significance and therefore have to be interpreted with caution.

A major focus of studies on animal travel is to look at how group movements are initiated and how cohesion is maintained (*Boinski & Garber, 2000*; *Fichtel & Manser, 2010*), yet close to nothing is known about how group movement and cohesion are regulated once individuals are on their way. While it is known that vocalizations can play a role in group cohesion (e.g., *Robinson, 1981*; *Cheney, Seyfarth & Palombit, 1996*; *Fischer et al., 2001*; *Trillmich, Fichtel & Kappeler, 2004*; *Braune, Schmidt & Zimmermann, 2005*; *Ramos-Fernández, 2005*) the results of our study suggest that individual movement patterns, i.e., changes in speed and direction, need to be addressed as a potential proximate mechanism as to how groups, which are made up of individuals, achieve and regulate cohesion (see also *da Cunha & Byrne, 2009*).

We propose that acceleration and deceleration of movements and adjustment of direction–in addition to initiating–is a domain that requires communication and our current findings support this view. In fact, fine-tuned regulation of group movements may be a common, hitherto largely overlooked, mechanism that is crucial in many group living species that depend on cohesion and occupy large home ranges. As such, vocally mediated movement regulation may be the default mechanism on a continuous scale with the more conspicuous initiation and termination of movements at the extremes.

Our hypothesis is that species that are constantly on the move, such as sooty mangabeys and other scramble foragers, may benefit specifically from a communication system that enables individuals to continuously regulate group movements as opposed to species for which group movements occur in discrete bouts for example to exploit clumped food resources (*Kinnaird & O'Brien, 2000*).

We might even expect that within-species variation exists as to what a coordination signal may mean. For example, frugivorous species are likely to travel in bouts during periods of high fruit availability and switch to continuous travelling in periods when food sources are dispersed, which is likely to exert different evolutionary pressures on a communication system. Sooty mangabeys exhibit such flexibility in their feeding ecology (*Bergmüller, 1998*, see also *Janmaat, Byrne & Zuberbühler (2006)*). Our data collection took place when food sources were dispersed, and it will be interesting to see how movements in this species are coordinated when resources are clumped (*Janmaat, Byrne & Zuberbühler, 2006*).

Our results also indicate that associations with other primate species and their vocalizations influence mangabey movements. Poly-specific associations among Taï monkeys are common (*McGraw, Zuberbühler & Noë, 2007*), offering mutually increased predator detection (*Bergmüller, 1998*; *McGraw & Bshary, 2002*). The cost/benefit ratio of these associations is high, given that they do not lead to increased food competition. Yet again, most data on communicative mechanisms so far demonstrate how associations are formed, rather than maintained and regulated (*Heymann, 2011*). Our results suggest that mangabeys adapt and coordinate their movements according to the presence and vocalizations of other primate species. These results suggest that the underlying

communicative mechanisms of interspecific movement coordination may be more complex than previously thought.

In sum, our results represent an example of a potential vocal mechanism by which movements of individuals can be influenced and, by extension, groups can be coordinated. Our results suggest a complex picture of how movement patterns and vocalizations, both within and between species, are interlinked and mediated by the immediate social environment. Future playback experiments will elucidate whether the relationships we suggest are indeed of a causal nature.

## ACKNOWLEDGEMENTS

We thank Richard Peho and Patterson for assistance in data collection, Anderson Bitty for logistic support and Julie Duboscq and the members of the Comparative Cognition Lab for discussions. We also wish to thank two reviewers and the editor for their constructive comments.

### Funding

This study was funded by the European Research Council (FP7/2007–2013, grant number 283871). The funders had no role in study design, data collection and analysis, decision to publish, or preparation of the manuscript.

### Grant Disclosures

The following grant information was disclosed by the authors:
European Research Council (FP7/2007–2013): 283871.

### Competing Interests

The authors declare that they have no competing interests.

### Author Contributions

- Christof Neumann conceived and designed the experiments, performed the experiments, analyzed the data, contributed reagents/materials/analysis tools, wrote the paper, prepared figures and/or tables, reviewed drafts of the paper.
- Klaus Zuberbühler conceived and designed the experiments, contributed reagents/materials/analysis tools, wrote the paper, reviewed drafts of the paper.

### Animal Ethics

The following information was supplied relating to ethical approvals (i.e., approving body and any reference numbers):

This study was entirely observational and adhered to the legal requirements of Côte d'Ivoire and Switzerland, as well as to the Animal Behavior Society Guidelines for the Use of Animals in Research.

## Field Study Permissions

The following information was supplied relating to field study approvals (i.e., approving body and any reference numbers):

Research permissions were granted by the Ministère de la Recherche Scientifique et Technique of Côte d'Ivoire.

## Data Deposition

The raw data has been supplied as Supplemental Dataset Files.

## Supplemental Information

Supplemental information for this article can be found online at http://dx.doi.org/10.7717/peerj.2298#supplemental-information.

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
