# Peer review of "Vocal correlates of individual sooty mangabey travel speed and direction"

_PeerJ, doi:10.7717/peerj.2298_

## Round 0.1 · original submission · Minor Revisions

Both referees and myself share the opinion that your ms is of interest for PeerJ and will deserve publication after minor/major revision.

Please include in your rebuttal letter how you have revised your MS by addressing all the points outlined by the two referees (see also letters of the referees) and myself.

1. According to my opinion and referee 1, it was not shown in your MS that vocalizations do really regulate travel speed and direction, you showed that vocal activity of the subgroup, but not the individual, may be linked to travel speed and direction. Thus, please take this into account and tone down Title and Content accordingly.

2. Referee 2 and myself think that you have to provide readers with a broader scope on the role of vocalizations in coordinating social cohesion and coordination. Thus, please include in the Introduction/Discussion the current literature on the role of vocalizations in coordinating and maintaining group cohesion ( e.g. not only anthropoid primates are doing so, but also basal primates with varying social systems (e.g. brown lemurs, woolly lemurs, mouse lemurs).

3. Methods need revision to make the paper understandable for the general audience.
Please address in the Methods
-how you identifed the respective mangabey groups/ subgroups/individuals,
- according to which criteria vocalizations were assigned to the respective vocalizing individual/ monkey or non-monkey species and to the respective sender.
-according to which critria vocalizations were assigned to a grunt, twitter etc.

Reviewer 1 ·

Basic reporting

The article reports a study that aimed to investigate the vocal regulation of individual sooty mangabey travel speed and direction. This is a interesting topic, because as the authors state in the introduction, not much is known about the role of vocalizations to moderate movement speed and changes in direction in social living mammals.
However, in my option, the article has some shortcomings that need to be adressed prior to publication. Most importantly, the paper did not entirely convince me that sooty mangabeys vocalisations function to regulate travel speed. The results are diverse and influenced by several aspects like group size and the presence of poly-specific associations, and potentially infraspecific social interactions, maybe sex and age rations, etc. This makes the paper very speculative in general, and the discussion to me does not propoerly address why the results (reporting opposite effects) depended so strongly on group size and the number of other primate species around. In particular, since aspects like composition of the subgroup (age and sex) as well as social interactions have not been taken into consideration. So as I understand the paper, it might be possible to somehow correlate the vocal activity with travel speed, but we do not know whether vocalisations per se regulate travel speed. Might it be possible in fact be social interactions and behaviour regulate travel speed, and that vocalisations are merely a means to measure that?

In addition, sometimes I found the paper very hard to read, with a lot of long and complicated sentences. This might be more of a personel opinion, but for example the sentence in the abstract line 10-13 is very complex and long. There is a lot of important information squeezed into one sentence.

Experimental design

The experimental design is sound, but I would ask for clarifications in the method section (see detail below).
line 70: please give sex ratio.
line 73: how have vocalisation types been identified? by hearing (by the observer), or by checking the (visually) spectrogram later in the lab. That is also not clear after reading the supplementals.
Otherwise the experimental setup is sound, and what is missing seems to be explained in the supplemental material. Although it might be worse to mention the behavioural categories also in the main paper.

Validity of the findings

The findings are presented in a reasonable way. But it becomes clear that many aspects, besides the vocalisations, influence the travel speed and direction changes. in particular, the fact that the opposite effects are found in relation to subgroup size, for example, really need to be addressed in the discussion part. In my opinion, conclusions need to be toned town, and it should be stated that due to the complex nature of the travel behaviour, more research will be needed in oder to draw final conclusions.
I am not convinced that grunts and twitters per se regulate the travel speed, as presented as the main finding of this paper.

line 176. please define in the main paper what is a small group, and what is large subgroup. Do you have any information of the sex ratio or the age group (infant versus adults) of the members of the subgroup?

line 210: if results are not statistically significant, you should interpret with caution, not with "some caution". There might be a tendency, but you have so many factor influencing these results, that nonsignificant findings should really be presented as preliminary findings that need further investigation.

244: tone down conclusion.

Additional comments

No comments

Reviewer 2 ·

Basic reporting

In this ms the authors showed that mangabeys use vocalizations to coordinate group movements by moderating travel speed and individuals travel direction. In one group of wild mangabeys 16 individuals were recorded for 14.6 hrs using animal focal recordings and movements of the focal animal were tracked simultaneously with a GPS. Vocalization rates of the subgroup an individual was part of, but not vocalization rate of the focal animal, predicted future travel speed and to a certain degree changes in travel direction. Thus, vocalizations in primates are not only used to initiate group movements but also to coordinate group movements.
Many other studies in this research examined if and how vocalizations are used to initiate group movements but not how vocalizations are used to coordinate travel speed and travel directions. Therefore, this ms addresses a new and interesting approach and should be of general interest for readers of PeerJ. The current knowledge on this topic is well summarized and the ms is written well. I have a few comments that I will outline in detail below.

Introduction
Please incorporate in the Introduction the current literature on the function of close calls because this research has shown that vocalizations in many species are used to maintain group cohesion. The authors address this aspect briefly in the Discussion but I think it is important to address it already in the introduction to present a more complete overview on the theoretical background.

Line 80: present already here the sample size, also indicate how many hours each individual was recorded
Line 81: explain what you mean with audible, this is not really a good term.
Line 88: if you measured the travel distance within 5 minutes and incorporated the distance in the models you should also call it distance. Explain why you included distance and not speed in the model.
Line 204: please, explain briefly what do you mean with “...serve as food calls similar to those described in chimpanzees” Not every ready might be familiar with this study and the function of food calls is not straightforward and has been discussed quite a lot in the literature.

Experimental design

no comments

Validity of the findings

Although the samples size is rater small (14.6 of focal recordings) the methods are in principle flawless.

---

## Round 0.2 · accepted · Accept

The authors addressed all the concerns of the referees and myself nicely and the submission can be accepted and processed for publication. It will present an interesting contribution to the role of vocalizations in primate social systems.

Reviewer 1 ·

Basic reporting

"no comments"

Experimental design

No comments

Validity of the findings

No comments

Additional comments

I appreciate that the authors addressed all my concerns. I now like the paper very much and belief that it will be a great and interesting contribution to the scientific literature.